# Short communication: Modelling competing effects of cooling rate, grain size and radiation damage in low temperature thermochronometers

David M. Whipp[1]★, Dawn A. Kellett[2]★, Isabelle Coutand[3], Richard A. Ketcham[4]

[1]Department of Geosciences and Geography, University of Helsinki, FI-00014 University of Helsinki, Finland, ORCiD ID: 0000-0002-3820-6886
[2]Geological Survey of Canada - Atlantic, Natural Resources Canada, Dartmouth, B2Y 4A2, Canada, ORCiD ID: 0000-0002-4558-4703
[3]Department of Earth and Environmental Sciences, Dalhousie University, Halifax, B3H 4R2, Canada,
ORCiD ID:
[4]Department of Geological Sciences, Jackson School of Geoscience, University of Texas, Austin, TX 78712, USA, ORCiD ID: 0000-0002-2748-0409

*Correspondence to*: Dawn A. Kellett (dawn.kellett@canada.ca)

★ These authors contributed equally to this work.

**Abstract.** Low temperature multi-thermochronometry, in which the (U-Th)/He and fission track methods are applied to minerals such as zircon and apatite, is a valuable approach for documenting rock cooling histories and relating them to geological processes. Here we explore the behaviours of two of the most commonly applied low temperature thermochronometers, (U-Th)/He in zircon (ZHe) and
apatite (AHe), and directly compare against the apatite fission track (AFT) thermochronometer for different forward-modelled cooling scenarios. We consider the impacts that common variations in effective spherical radius (ESR) and effective Uranium concentration (eU) may have on cooling ages and closure temperatures under a range of different cooling rates. This exercise highlights different scenarios under which typical age relationships between these thermochronometers (ZHe > AFT >
AHe) are expected to collapse, or partially to fully invert. We anticipate that these predictions and the associated software we provide will be a useful tool for teaching, planning low temperature multi-thermochronometry studies, and for continued exploration of the relative behaviours of these thermochronometers in the temperature-time space through forward models.

## 1 Introduction

Low temperature multi-thermochronometry, particularly involving the incorporation of both (U-Th)/He and fission track datasets, represents the state of the art for developing temperature-time (T-t) evolutions for rocks in the upper continental crust. Track length distributions in fission track thermochronology and effective Uranium (eU; calculated as [U]+0.238[Th])-age relationships in (U-Th)/He

thermochronology (Cooperdock et al., 2019) together have the potential to provide highly detailed rock
T-t histories that can be used to interpret and reconstruct a diverse range of geological processes and
their rates, from long-term landscape evolution of Precambrian shields (e.g. Lorencak et al., 2004;
Danišík et al., 2008) to rifting (e.g. Cogné et al., 2011; Ricketts et al., 2016) to orogenic construction
and collapse (e.g. Thomson and Ring, 2006; Coutand et al., 2014; Toraman et al., 2014). The
diffusion/annealing models for these thermochronometric systems, especially for the common accessory
minerals zircon and apatite, are fairly well accepted across the scientific community, and embedded into
widely-used thermal modeling softwares such as *HeFTy* (Ketcham, 2005) and *QTQt* (Gallagher, 2012),
as well as thermokinematic models such as *Pecube* (Braun, 2003).

While it has become common practice to input measured (U-Th)/He and fission track data into thermal
modelling software to invert for best-fit thermal histories, this type of application and interpretative
products can obscure visualization of the complex relationships that exist between internal (e.g., eU,
grain size, mineral chemistry) and external (thermodynamical effects of the various and competing
geological processes that lead to changes in rock T) parameters (or factors) controlling measured
thermochronometric ages. Classical plots of closure temperature vs. cooling rate, in which the
relationships for mineral-specific thermochronometers form a stack of near-parallel curves (e.g., Fig. 1
of Reiners and Brandon, 2006), are widely cited in courses and the literature, and often form the starting
point for discussing the significance of low temperature thermochronological datasets. However, those
plots seldom include the age and closure temperature effects in broadly accepted He diffusion models
that incorporate crystal damage and annealing (Flowers et al., 2009; Guenthner et al., 2013; Guenthner
2021). Forward modelling tools (including *HeFTy* and *QTQt*) are well-suited for exploring parameters
such as grain size and eU, because additional complicating factors that apply to empirical datasets, such
as chemical zoning or unexplained age dispersion, can be ignored, and because thermal histories are
user-defined rather than non-unique unknowns. However, batch-processing hundreds to thousands of
forward models to evaluate how broad ranges of input parameters affect predicted ages or closure
temperatures can be tedious. Here, we have designed a simple forward model software to examine
differences in predicted thermochronometer ages and closure temperatures with a particular focus on
comparing (U-Th)/He zircon and apatite systems (hereafter ZHe and AHe, respectively) to the apatite
fission track (AFT) system. Our goal is to explore and compare the range of behaviours of these
different systems that could be expected for different grain sizes and eU concentrations by generating
thermochronometric datasets for a wide range of linear cooling rates. The plots and associated code we
provide are useful interpretive tools for designing multi-thermochronometric studies, and for
conceptualizing expected thermochronometer behaviours under various geological conditions.

## 2 Predicting thermochronometer ages and closure temperatures

We used existing thermochronometer age prediction software to predict AHe, AFT, and ZHe
thermochronometer ages and effective closure temperatures for a range of cooling rates, eU
concentrations, and grain radii. Rather than calculating thermal histories using a heat transfer model, we
generated synthetic linear cooling histories with cooling from 350 to 0 °C at constant rates of 0.1 - 100

°C/Myr (Figure 1). This approach allows exploration of the effects of a wide range of plausible cooling rates through the partial retention and partial annealing zones of all three thermochronometers. To
explore the effects of radiation damage on He diffusion, we considered ranges in eU concentration of 1 – 150 ppm for AHe and 1 – 4000 ppm for the ZHe system. These different ranges are intended to reflect typical eU values for natural apatite and zircon grains that could be the target for dating (e.g., Donelick et al., 2005; Cherniak and Watson, 2003). Finally, we varied effective spherical radius (ESR) from 40 – 100 μm for both zircon and apatite, as an estimate of the natural variation in ESR in dated minerals.
Note that these models do not consider zonation of the parent isotopes at this time, which can strongly impact both the alpha-ejection correction (Hourigan et al., 2005) and He diffusion behaviours (Gautheron et al., 2012).

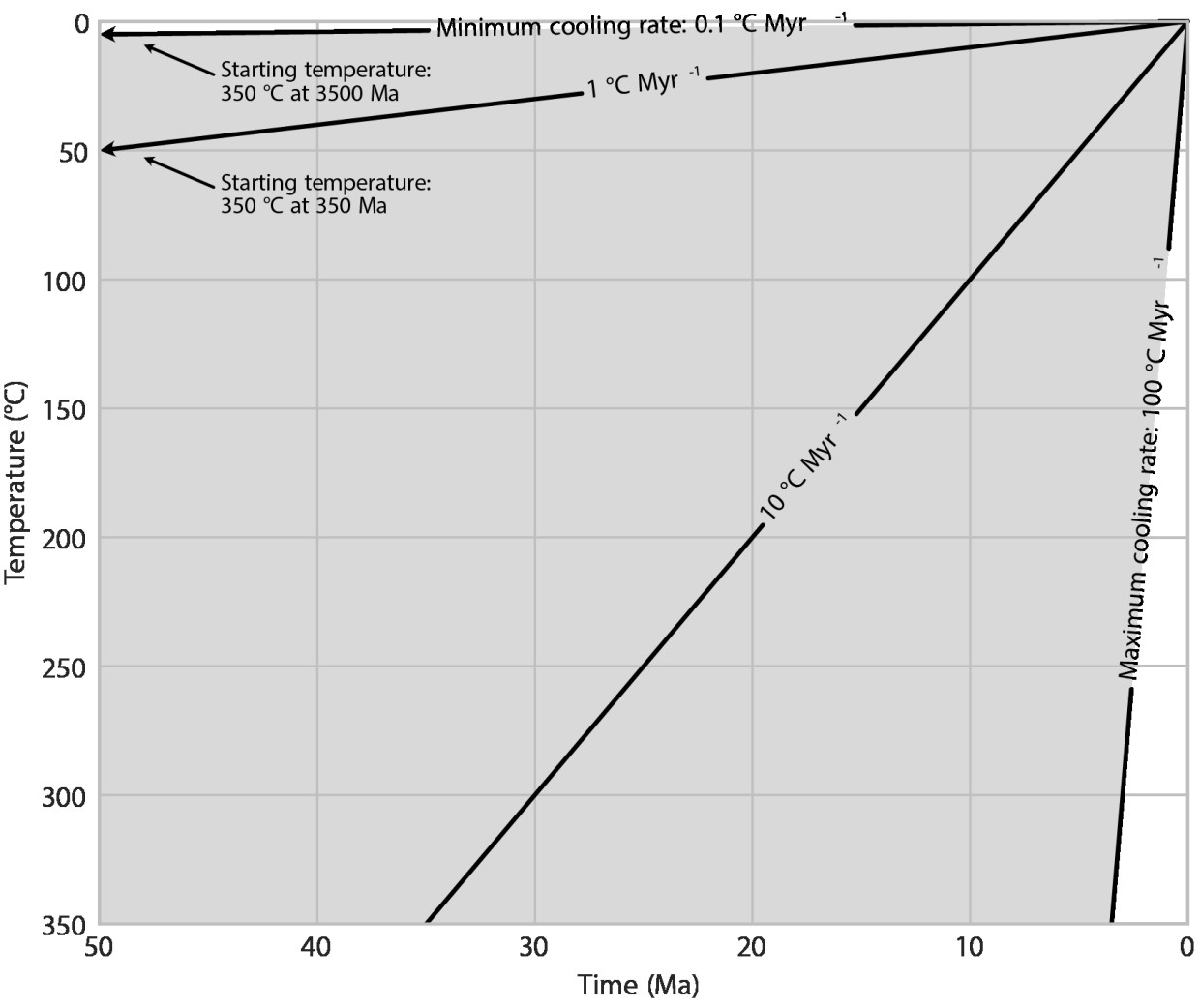


**Figure 1: Temperature-time plot showing the range of cooling histories (grey shaded area) used for thermochronometer age prediction. All scenarios start at 350 °C and cool to 0 °C at a constant rate. Note that the x-axis of the plot is truncated for**

readability. Fig. 2 shows results for a 10 °C/Myr cooling rate, Fig. 3 uses a 1 °C/Myr cooling rate, and Figs. 4 and 5 show results for the full shaded region in log space.

Using the predefined ranges in cooling rate, eU concentration, and ESR as inputs, we calculated thermochronometer ages and effective closure temperatures using the fission track annealing model of Ketcham et al. (1999) for AFT ages, and the radiation damage accumulation and annealing models of Flowers et al. (2009) and Guenthner et al. (2013) for simulating the effects of radiation damage on the predicted AHe and ZHe ages, respectively. The Cl content was set to 0 ppm for the fission-track age

prediction, and the (U-Th)/He age prediction software includes the effect of alpha ejection following Ketcham et al. (2011), which corrects the age based on the production of He from each parent isotope separately rather than correcting based on the age alone. For all cases, the effective closure temperature was estimated by reporting the temperature in the cooling history at the time of the predicted thermochronometer age.


  The software used comprises programs for predicting (U-Th)/He (Ketcham et al., 2018) and apatite fission-track (Ketcham et al., 2000, as implemented in Braun et al., 2012) closure temperatures and ages written by Richard Ketcham in the C and C++ programming languages, and new scripts written in the Python language for producing the cooling histories and plots. The software is all open source and

details about how to use the software and its licensing can be found in the Code availability section. We also provide an online interactive application (Jupyter notebook) that can be used to reproduce and customize versions of Figures 2-5 with nothing more than a web browser. Furthermore, in addition to the linear cooling histories presented here it is possible to define more complex thermal histories involving multi-stage cooling and reheating events, as well as export predicted AFT length

distributions. Details about how to use and customize the software are available in the Code availability section and code description in the software archive online.

## 3 Exploring the multi-thermochronometry space

### 3.1 Contrasting He behaviour in apatite and zircon under cooling scenarios

  He is produced in apatite and zircon primarily via alpha decay of U and Th, and its mobility

(loss/retention) in apatite and zircon forms the basis for AHe and ZHe thermochronometers, which are broadly applied by the Earth Sciences research community to determine temperature-time (T-t) points or paths for analysed rock samples. He mobility occurs both via alpha ejection (the implantation of He produced during U and Th decay into neighbouring grains due to the long stopping distance of the alpha particle), which is a function of the grain size and geometry (Meesters and Dunai, 2002a), and via

thermally-controlled volume diffusion, which is also sensitive to grain size and geometry (e.g., Reiners and Farley, 2001). Consequently, grain size and geometry are critical parameters in modeling thermal histories based on He dating. These are typically quantified using the grain equivalent spherical radius (ESR), based on the observation that isothermal outgassing of apatite well fits a spherical diffusion model, and that the spherical diffusion model reproduces diffusion results for more accurate geometries

such as the finite cylinder (Wolf et al., 1996; Meesters and Dunai, 2002b). However, He diffusion

behaviour in both apatite and zircon is also dependent on the progressive accumulation of internal crystal damage caused by alpha decay. Crystal damage occurs at a rate determined by the eU concentration in a crystal, and is thought to anneal in a similar way, and under somewhat similar thermal conditions to those needed for annealing of fission tracks (Flowers et al., 2009; Guenthner et
al., 2013; Guenthner, 2021). Consequently, both the pre-He retention thermal history and the chemistry of dated crystals (which together determine how much crystal damage has accumulated) are essential inputs for modelling He diffusional behaviour and determining grain specific AHe and ZHe closure temperatures. He diffusivity in apatite has been found to generally decrease with greater accumulated alpha damage, such that more damaged grains are more retentive and have higher AHe closure
temperatures (Shuster 2006; Shuster et al., 2009). Zircon commonly incorporates significantly more U (and hence eU) into its structure compared to apatite, with eU concentrations of 100-1000 ppm being typical and eU >4000 ppm being not uncommon (compared to more typical eU concentrations of 1-100 ppm in apatite). Zircon annealing temperatures are higher than for apatite, and zircon is also more resistant to geological cycling than apatite. Thus, the potential for accumulating radiation damage is
much higher in zircon compared to apatite. At low to intermediate levels of alpha decay-induced crystal damage, He diffusivity in zircon decreases with greater accumulated alpha damage, but at high levels of damage, He diffusivity increases significantly (Guenthner et al., 2013). Consequently, possible closure temperatures for the ZHe system show a much larger range than for AHe (e.g., Ault et al., 2019). Finally, we note that since annealing of fission tracks in apatite is not subject to volume diffusion, AFT
ages are not influenced by either apatite grain size or eU concentration (e.g. Kohn et al., 2009).

In the modelling presented here, we explore these relationships in the context of simple linear cooling histories, starting at high temperature in the past and cooling continuously until the present day. An important consideration in such histories is that radiation damage begins to accumulate before any
helium is retained, in both the apatite and zircon systems. In this framework, slower cooling rates will permit accumulation of more damage, and in the result below we see multiple manifestations of the interplay between cooling rate and diffusivity evolution. To allow the opportunity for radiation damage to accumulate in both apatite (<200 ˚C) and zircon (<350 ˚C), all models are started at 350 ˚C.

First, we investigated the extent of grain size and eU concentration controls on He diffusion (and resulting (U-Th)/He closure temperatures) in zircon and apatite for a constant cooling rate of 10 °C/Myr and typical ESR ranges (40-100 μm) and eU concentrations (1-150 ppm for apatite, and 1-4000 ppm for zircon; Fig. 2). This cooling history results in a total model run time of 35 Myr. At this cooling rate and timescale, the AHe (Fig. 2a,b) and ZHe (Fig. 2c,d) thermochronometers show contrasting relationships.
AHe cooling ages are strongly positively correlated with ESR, with smaller grains having younger cooling ages and lower closure temperatures, and larger grains having older cooling ages and higher closure temperatures (Fig. 2a, b). The AHe cooling ages are much less sensitive to variations in eU concentration, although still positively correlated. These plots show that, over this relatively short timescale, alpha damage exerts little influence on He diffusion (or diffusional behaviour) in apatite, and
AHe closure temperature varies by less than 15 °C across these scenarios. However, ZHe cooling ages show the opposite relationship (Fig. 2c, d). The higher natural range in zircon eU concentrations, and the resultant damage to the zircon crystal from those higher dosages of alpha decay, controls the

diffusion behaviour of He in zircon even in this relatively rapid cooling scenario. Consequently, for the same cooling history as that modelled for apatite, ZHe cooling ages and closure temperatures are strongly positively correlated with eU concentration, relatively insensitive to ESR, and ZHe closure temperature varies >100 °C (the full range of closure temperature in Fig. 2d is 73.5-192.9 °C).

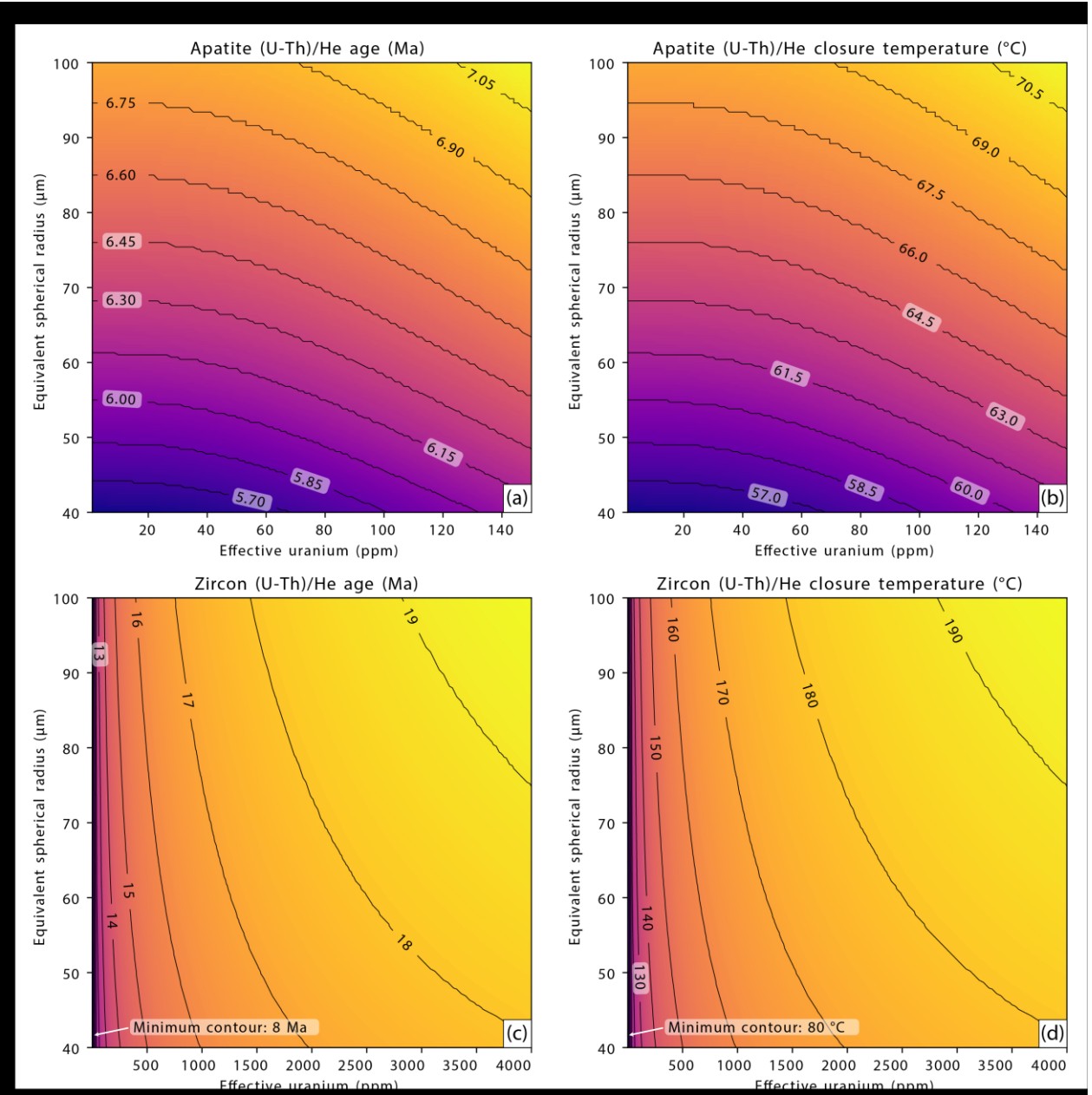

**Figure 2: Contoured model (U-Th)/He cooling ages (a) and closure temperatures (b) for apatite, and model (U-Th)/He cooling ages (c) and closure temperatures (d) for zircon of different effective spherical radii and eU concentrations (ppm). All panels are calculated for cooling from 350 °C to 0 °C at a constant rate of 10 °C/Myr. The plots comprise predicted ages and closure temperatures for 10,201 forward models.**

While the 10 °C/Myr cooling rate applied in Figure 2 could represent active orogenic settings, slower
cooling rates are also common to many geological environments. To compare the effect of an order of magnitude slower cooling on He diffusion, we have applied all the same parameters as in the 10 °C/Myr scenario to a 1 °C/Myr constant cooling rate, equating to a model run time of 350 Myr (Fig. 3). The primary difference in the behavior of He in apatite under slower cooling is that AHe ages and closure temperatures correlate much more strongly with eU concentration than with ESR, resulting in ~30 °C
variability in closure temperature over the range of eU concentration considered (Fig. 3b) and variation in AHe age of up to 30 Myr (Fig. 3a). The slower cooling provides a greater period of time for accumulation of alpha decay-induced crystal damage. In an empirical study, such a cooling scenario could be expected to produce statistically-significant positive age-eU correlations. The ZHe system in this scenario continues to be insensitive to ESR, and while it is highly sensitive to eU concentration for
values <500 ppm, it is quite insensitive to eU concentration for values >500 ppm (Fig. 3c,d). Thus, there is an eU concentration threshold above which zircons may not show an age-eU relationship. This threshold could be important to recognize when interpreting the significance of zircon age-eU plots, at least in the context of cooling-only histories.

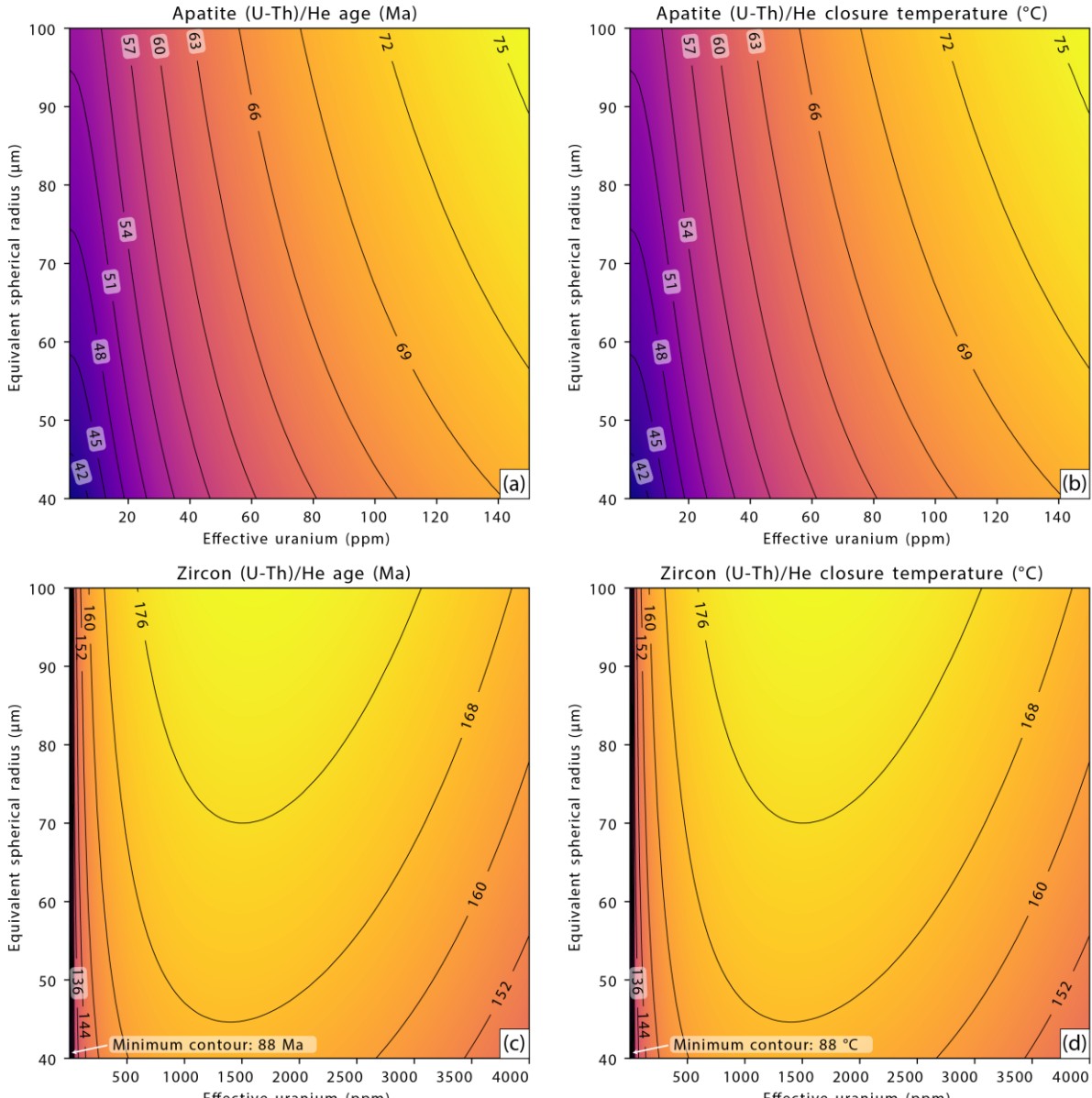

**Figure 3: Contoured model (U-Th)/He cooling ages (a) and closure temperatures (b) for apatite, and model (U-Th)/He cooling ages (c) and closure temperatures (d) for zircon of different effective spherical radii and eU concentrations (ppm). All panels are calculated for cooling from 350 °C to 0 °C at a constant rate of 1 °C/Myr. Note that the similarities between the left and right panels is because the ages and closure temperatures are expected to have the same value for a cooling rate of 1 °C/Myr. The plots comprise predicted ages and closure temperatures for 10,201 forward models.**

In contrast to Figures 2 and 3, where only a single cooling rate was applied, we next explore the influence of the intra-grain parameters (grain size, eU) on closure temperature for a wide range of geologically plausible cooling rates (0.1-100 °C/Myr; Fig. 4). We first fixed the eU at illustrative values of 10 ppm for apatite and 100 ppm for zircon while varying ESR (Fig. 4a, c), and then fixed ESR at

illustrative values of 45 μm for apatite and 60 μm for zircon while varying eU (Fig. 4b, d). In this parameter space, closure temperatures for AHe range from ~30-85 °C, closure temperatures for ZHe vary from ~0-185 °C, and closure temperature relationships for AHe and ZHe are again strongly contrasting. For eU of 10 ppm, AHe closure temperature is negatively correlated with cooling rate at slow cooling rates, but undergoes an inflection at ~0.5-1 °C/Myr beyond which closure temperatures are

positively correlated with cooling rate at faster cooling rates, with variations in eU maintaining an overall positive correlation with closure temperature at all cooling rates (Fig. 4a). For eU of 100 ppm, ZHe closure temperatures, in contrast, are negatively correlated with cooling rate, with a subtle inflection inverting this relationship at very slow cooling rates approaching 0.1 °C/Myr. As with apatite, zircon ESR variations show a positive correlation with closure temperature under the full range of

cooling rates (Fig. 4c).

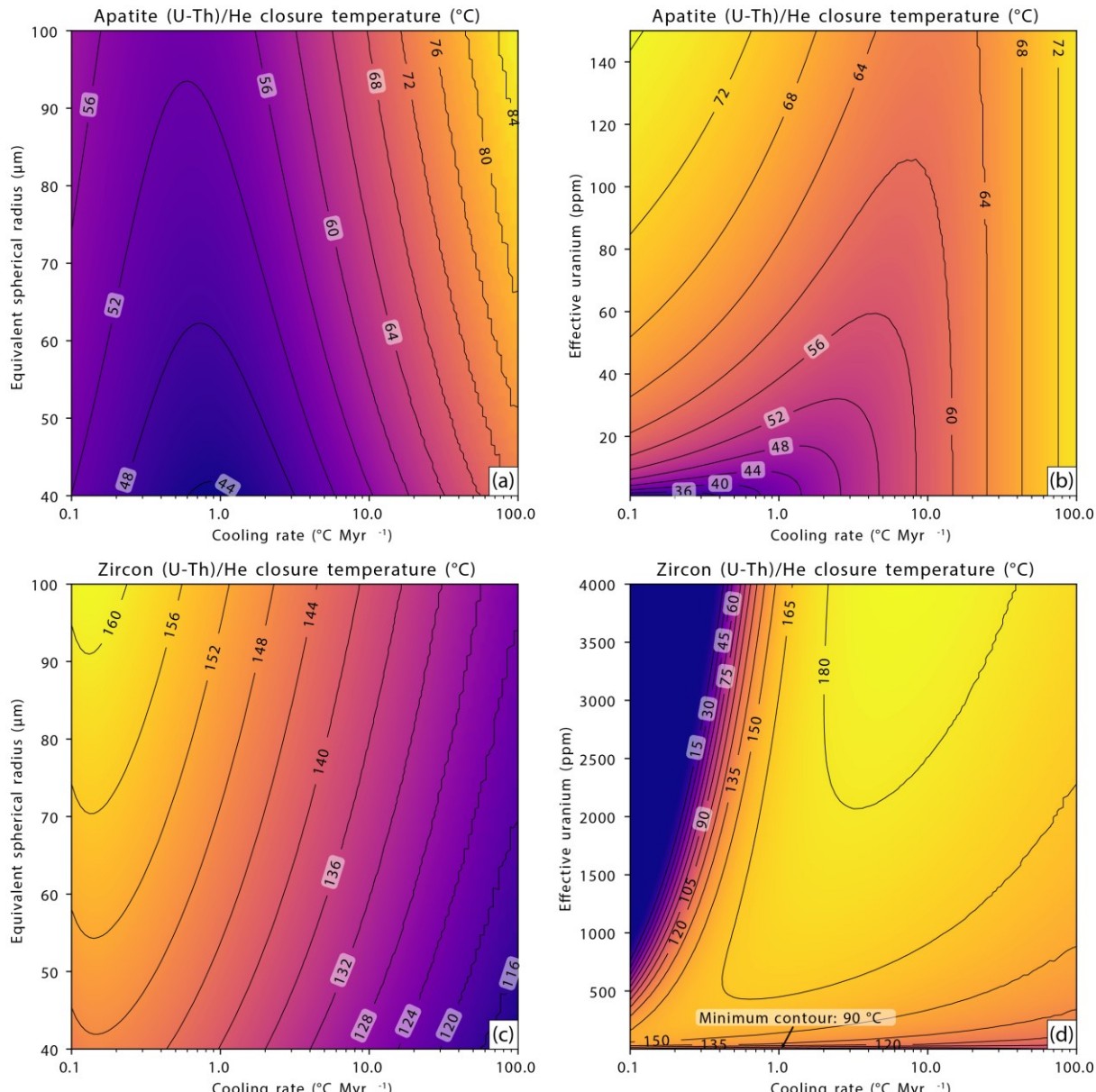

**Figure 4: Contoured closure temperatures for the apatite and zircon (U-Th)/He systems as functions of cooling rate, effective spherical radius, and eU concentration. (a) Apatite: eU is fixed at 10 ppm; (b) apatite: ESR is fixed at 45 µm; (c) zircon: eU is fixed at 100 ppm; (d) zircon: ESR is fixed at 60 µm  The plots comprise predicted closure temperatures for 20,402 forward models and each model applies a constant cooling rate between 0.1-100 °C/Myr.**


For a fixed ESR of 45 µm, the AHe system shows closure temperatures that are positively correlated with eU and negatively correlated with cooling rate at slow cooling rates, but undergoes an inflection between 1-10 °C/Myr beyond which closure temperatures are positively correlated with cooling rate,

and insensitive to eU (Fig. 4b). For the same cooling rate range, a fixed ESR of 60 μm, and range of zircon eU values of 1-4000 ppm, the ZHe system shows closure temperatures that are strongly positively correlated to cooling rate, and strongly negatively correlated to eU concentration for slow cooling rates (≤1 °C/Myr), except for low eU (<50 ppm), such that the closure temperature approaches 0 °C over a significant area of the plot space in which damage accumulation is high (high eU, slow

cooling rate; Fig. 4d). This system also shows an inflection, in the vicinity of 0.1-2 °C/Myr, beyond which ZHe closure temperatures are weakly positively correlated to eU, and actually negatively correlated with cooling rate. In other words, fast cooling rates are expected to produce lower ZHe closure temperatures than intermediate cooling rates. Although this result seems potentially contrary to thermal diffusion systems in which faster cooling rates generally result in higher closure temperatures

(e.g. Fig. 4b), in the case of zircon, the faster cooling scenarios also result in the most pristine crystals because of the progressively shorter time to accumulate alpha damage (low eU, fast cooling rate). Moderate doses of alpha damage are known to decrease diffusivity compared to pristine to low alpha damage dose crystals (Guenthner et al., 2013). The implication is that zircon with low eU under fast cooling rates are more sensitive to the degree of alpha damage than to cooling rate, at least in cooling-

only scenarios. Comparison of these four plots (Fig. 4) indicates that ZHe and AHe closure temperatures are not only expected to vary significantly under different cooling rate scenarios, but also that they do not track together, meaning that some conditions simultaneously favour higher AHe and lower ZHe closure temperatures and vice versa. We explore this outcome in more detail in the following section. Differences in closure temperature behaviour should also, of course, be expected for zircon

from neighbouring but different rock types that share a cooling history but may have quite different ranges in eU concentration, not to mention within samples for which individual grains show large ranges in eU, as is common in detrital samples. Likewise for intra-sample ranges in ESR for apatite or zircon, although these differences are likely more subtle.

**3.2 The multi-thermochronometry space of ZHe, AFT and AHe**

The relationships between ZHe, AFT and AHe have commonly been summarized as stacked semi-parallel curves in a plot of closure temperature vs. cooling rate (e.g., Reiners and Brandon, 2006), or as having progressively lower closure T ranges in lists of widely applied thermochronometers. However, it is apparent in the above plots that 'typical' zircon and apatite are expected to have contrasting He

diffusion behaviours under different cooling scenarios. In Figure 5, we provide a visualization of how these different behaviours at high vs. low cooling rate are expected to produce contrasting cooling age and closure temperature relationships among the ZHe, AFT and AHe thermochronometers. In each plot pair, we have predicted ages and closure temperatures of the three thermochronometers for constant cooling rates of 0.1-100 °C/Myr, corresponding to cooling from 350 °C in 3500 to 3.5 Myr, respectively

(Figs. 1, 5). For simplicity, we varied eU in the stacked plots from low (apatite = 1.0 ppm; zircon = 10 ppm), to intermediate (apatite = 10 ppm; zircon = 100 ppm), to high (apatite = 200 ppm; zircon = 1000 ppm) eU concentrations. We note that in nature, a rock with high eU zircon may not necessarily have high eU apatite, and vice versa, but present the plots in this way for simplicity. In all plots, AFT is unaffected by eU, and shows near-linear relationships between predicted age and cooling rate, and

closure temperature and cooling rate in log-log and semi-log space, respectively. To estimate the conditions in which measured ages may differ between systems including their measurement uncertainties, we have predicted cooling ages as age swaths with "typical" uncertainties of 10% for AHe and ZHe, and 20% for AFT.

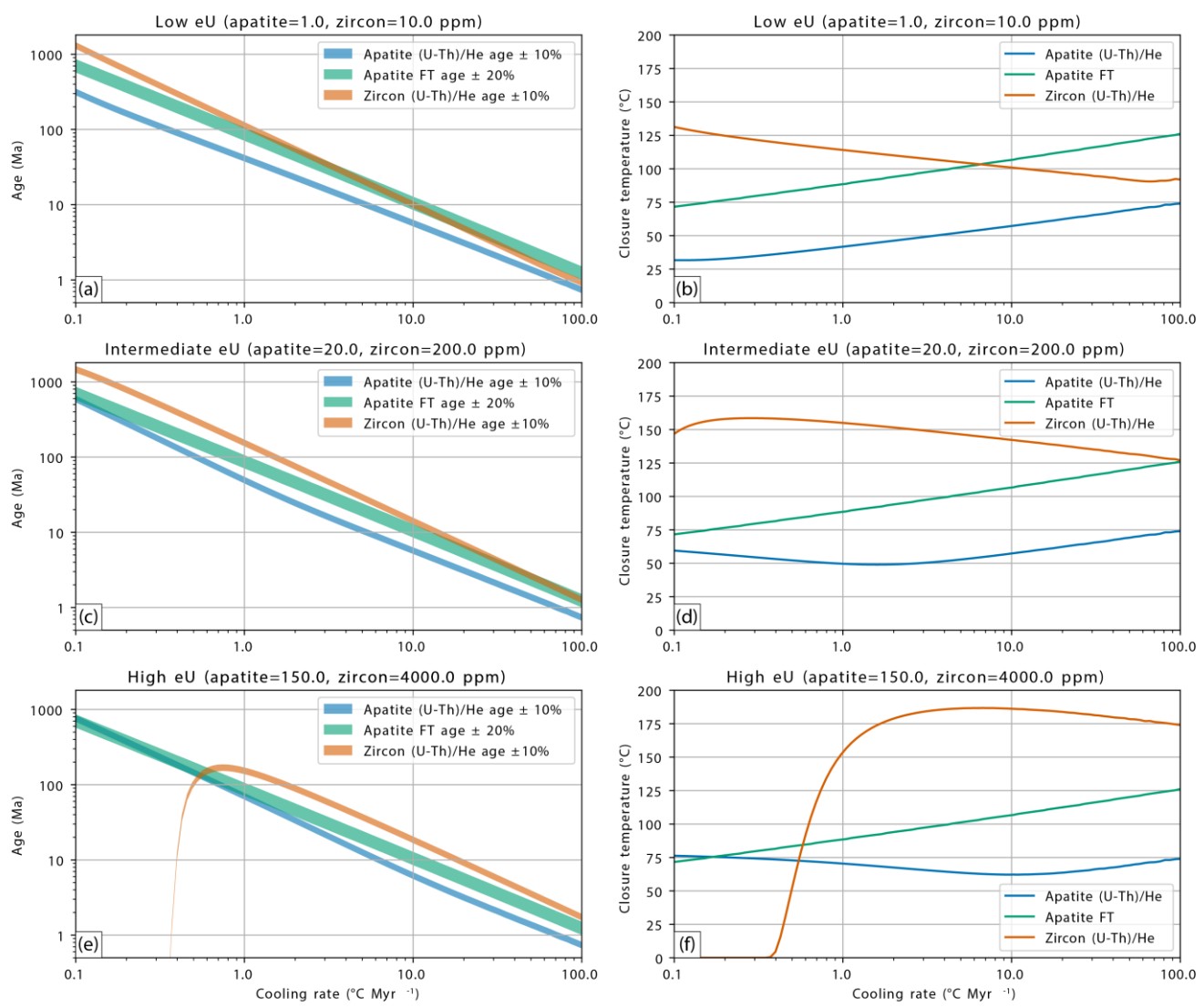


**Figure 5: Predicted thermochronometer ages (left) and closure/annealing temperatures (right) for low (top), intermediate (middle), and high (bottom) eU concentrations, as a function of cooling rate. The plots comprise predicted ages and closure temperatures for 303 forward models and each model applies a constant cooling rates between 0.1-100 °C/Myr. The coloured swaths for the predicted ages (a, c, e) indicate the mean age plus or minus the indicated percent uncertainty.**

The AHe cooling ages and closure temperatures are non-linear in this parameter space (Fig. 5), being uniformly younger/lower than AFT, except for high eU apatite at very slow cooling rates (Fig. 5e, f), for which AHe cooling ages and closure temperatures could be slightly older and higher than AFT. The

lowest AHe closure temperatures are expected for slow cooling rates of 0.1-2 °C/myr and low-intermediate eU (Fig. 5b, d), while at high eU, the lowest AHe closure temperatures are expected for
intermediate-fast cooling rates of 10-20 °C/Myr (Fig. 5f). ZHe cooling ages and closure temperatures
are also generally non-linear. At slower cooling rates (<10 °C/Myr for low eU; <50 °C/Myr for
intermediate eU), ZHe cooling ages and closure temperatures are expected to be older/higher than AFT,
while at faster cooling rates (>10 °C/Myr for low eU; >50 °C/Myr for intermediate eU), they are
younger/lower than AFT (Fig. 5a-d). The high eU scenario clearly shows the remarkable changes in He
diffusivity for high alpha doses in zircon, with a dramatic drop in cooling age and closure temperature
expected for cooling rates slower than ~1 °C/Myr. Thus, although under most of the parameter space
explored, cooling ages are expected to progressively decrease from ZHe to AFT to AHe, there are
conditions under which this relationship partially (AFT>ZHe for fast cooling rates, low/intermediate
zircon eU) or fully (AHe>AFT>ZHe for very slow cooling rates and high eU) inverts, even when
including large error bars for the calculated ages. These relationships may in part explain observations
from empirical studies. For example, AHe > AFT ages have been commonly reported from geologically
old (cratonic) regions (e.g., Hansen and Reiners, 2006; Danišík et al., 2008; Flowers and Kelley, 2011).
As shown in these plots, AHe and AFT ages are expected to converge and invert for high eU in apatite
and timescales >250 Ma (Fig. 5f). AHe > ZHe ages have also been reported from cratonic samples with
high-damage zircon (Johnson et al, 2017). The highest damage zircon simulated in the linear cooling
scenarios presented here is represented by slow cooling of high eU zircon (Fig. 5e, f). These plots form
a first-order guide to investigating the character of regional multi-thermochronometry datasets, and the
software we provide can be used by the reader to further explore expected relationships and time lags
between the chronometers under either constant, or multi-stage linear cooling and heating scenarios, as
well as other parameters such as apatite composition in the context of the AFT system.

## 4 Summary

The ZHe, AFT and AHe methods are commonly used together in samples to develop low temperature
thermal histories for rocks and regions. In this short communication, we have explored the range of
cooling age and corresponding closure temperature responses expected for ZHe and AHe, relative to the
AFT thermochronometry system, by exploring typical parameter ranges for these systems using simple
forward temperature-time models. We compared the relative effects of grain size and eU on ZHe and
AHe closure temperature and cooling age, and showed that under typical mineral-specific ranges of eU,
the ZHe system is highly sensitive to eU and comparatively insensitive to grain size, while the AHe
system is sensitive to grain size and less sensitive to eU. The complex relationships that the ZHe and
AHe systems exhibit with respect to eU and grain size result in contrasting relationships among the
three thermochronometers under different linear cooling scenarios, including convergence between the
thermochronometers, and even partial to full inversion of the typical ZHe > AFT > AHe age
relationship. The software available from this study provides a new tool to easily forward model multi-thermochronometry relationships, and complements the range of existing modelling software packages
for thermochronological research.

## Code availability

The version of the software used to produce the figures in this manuscript is available at
https://zenodo.org/record/6034068 . Updated versions of the software can be obtained from
https://github.com/HUGG/tcplotter. Users can create, customize, and save plots using an online
interactive version of the software available at
https://mybinder.org/v2/gh/HUGG/tcplotter/HEAD?urlpath=lab/tree/tcplotter.ipynb.

## Author contribution

All authors conceptualized this study. Whipp wrote the plotting scripts and produced the figures in
consultation with Kellett and Coutand, and with assistance from Ketcham. Kellett and Whipp wrote the
manuscript. Whipp, Kellett, Coutand, and Ketcham developed the discussion, and revised the
manuscript.

## Acknowledgements

This is Natural Resources Canada contribution #20210364.

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

Assets:

The version of the software used to produce the figures in this manuscript is available at https://zenodo.org/record/6034068 . Updated versions of the software can be obtained from https://github.com/HUGG/tcplotter.


Short summary:

Multi-thermochronometry, in which methods such as (U-Th)/He dating of zircon and apatite, and apatite fission-track dating are combined, is used to reconstruct rock thermal histories. Our ability to reconstruct thermal histories and interpret the geological significance of measured ages requires

modeling. Here we use forward models to explore grain size and chemistry effects on cooling ages and closure temperatures for the (U-Th)/He decay systems in apatite and zircon.