# Peer review of "Short communication: Modelling competing effects of cooling rate, grain size and radiation damage in low temperature thermochronometers"

_Geochronology, 2021_

## Author Response (AR1)

We greatly appreciate the suggested revisions and recommendations of the associate editor. Below we provide our responses and indicate any changes to the manuscript or code.

Thank you for responding to the reviewer comments on your contribution to GChron. To these comments, I would like to add a few remarks of my own. Please take these into consideration before submitting your revisions.

1. Your software is, essentially, a Python front end to some forward modelling functions that were lifted out of Richard Ketcham's HeFTy code. Please be more precise in the acknowledgments which, in their present form, consider 'algorithm' to be a synonym for 'source code'. This is not the case. It would be better to thank Dr. Ketcham for "sharing source code used for thermochronometer age prediction in HeFTy", or something like that.

We agree that we have understated the value of Richard Ketcham sharing his software for predicting thermochronometer ages. After some additional thought, we found it reasonable to invite him to be a co-author of the article, as he had not only shared his software, but made some modifications that helped us to be able to get started using it with our Python software. The paper has been edited accordingly to reflect this change in authorship. In addition, he proposed updating our models to begin at 350 °C to ensure all possible damage accumulation in zircon is captured. That is updated that throughout all text and plots accordingly – with minimal change to our observations and interpretations. Note also that Richard identified an incorrect initial track length value in the original FT code which resolved the few percent discrepancy between our and HeFTy AFT results. They are now within 0.3%.

2. Whereas HeFTy is famously user-friendly, your Python code is not. The fact that it requires a c/c++ compiler will turn away 95% of your potential readers. And the 5% who do know how to compile code on Linux could easily write a Python-wrapper for diffusion modelling themselves. If you want your paper to have a real impact, I would suggest that you put some effort into making your software easier to use by the average geologist. If creating a GUI is too much to ask, then I would suggest that you try and release a precompiled Windows (and Mac) version via pip, say. Alternatively, if you could translate Dr. Ketcham's code into Python, then you could release a platform-independent package this way. I will not set the user-friendliness of the software as a hard requirement for publication, but I would strongly recommend that you give this some thought nonetheless.

We thank the associate editor for this important consideration, and have made several steps towards making our software more user friendly. First, we have created an interactive online version of the code available in the form of a Jupyter notebook. We have included a link to this notebook in the software repository for the code that allows users to test out the code, make custom plots, and save the plots as files using nothing more than a web browser (see https://mybinder.org/v2/gh/HUGG/tcplotter/HEAD?urlpath=lab/tree/tcplotter.ipynb). In addition, we have created additional documentation and repackaged the code to be easier to install in Windows, macOS, or Linux. We are in the process of seeking approval to add the software, now known as tcplotter, to the conda-forge software repository and from which it would be installable with a single command. Unfortunately, this approval has not yet come.

3. In its present form, the manuscript describes the U-Th-He age system as a simple interplay between radiation-damage-controlled thermal diffusion and linear cooling, without mentioning the effect of alpha ejection. Alpha ejection has a first order effect on U-Th-He ages, and is independent of

temperature. The complex interplay between diffusion, grain-size and alpha-ejection is beautifully described in a pair of papers by Meesters and Dunai (2002, doi:10.1016/S0009-2541(01)00422-3 and doi:10.1016/S0009-2541(01)00423-5). I was surprised that these were not cited in your manuscript.

We agree that this was an omission in our manuscript. We now explicitly state that our code does not explore variations in alpha ejection correction due to zonation in parent isotopes. We have added/modified the following statements in the text:

In section 2: "Note that these models do not consider zonation of the parent isotopes at this time, which can strongly impact the alpha-ejection correction (Hourigan et al., 2005) and He diffusion behaviours (Gautheron et al., 2012)."

In section 3.1: "He mobility occurs both via alpha ejection (the implantation of He produced during U and Th decay into neighbouring grains due to the long stopping distance of the alpha particle), which is a function of the grain size and geometry (Meesters and Dunai, 2002a), and via thermally-controlled volume diffusion, which is also sensitive to grain size and geometry (e.g., Reiners and Farley, 2001). Consequently, grain size and geometry are critical parameters in modeling thermal histories based on He dating. These are typically quantified using the grain equivalent spherical radius (ESR), based on the observation that isothermal outgassing of apatite well fits a spherical diffusion model, and that the spherical diffusion model reproduces diffusion results for more accurate geometries such as the finite cylinder (Wolf et al., 1996; Meesters and Dunai, 2002b)."

According to lines 88-89: "the effective closure temperature was estimated by reporting the temperature in the cooling history at the time of the predicted thermochronometer age." Is the "predicted thermochronometer age" calculated with or without an alpha-ejection correction? And if the age was calculated with alpha ejection correction, then how was this done? Did you apply a continuous alpha correction, or a nominal correction? From my questions you can see that the meaning of a closure temperature is not so straightforward in the context of U-Th-He thermochronology.

The ages calculated using the Ketcham software do have an alpha-ejection correction applied to them, and we have now clarified this in the text (see below). The applied alpha-ejection correction is from Ketcham et al. (2011), and corrects the age based on the production of each parent isotope separately rather than correcting the age or He content based on the total measured He content as a whole. The revised text starting on line 103 reads:
"The Cl content was set to 0 ppm for the fission-track age prediction, and the (U-Th)/He age prediction software includes the effect of alpha ejection following Ketcham et al. (2011), which corrects the age based on the production of He from each parent isotope separately rather than correcting based on the age alone."

4. Meesters and Dunai (2002) also incorporate the effects of compositional zoning, which is another factor that your paper ignores. Tibor Dunai released a neat little computer program called DECOMP (doi:10.2138/rmg.2005.58.10) that allows the user to forward-model U-Th-He ages as a function of time and space. It is an excellent example of a user-friendly interface. But despite these qualities, DECOMP never got much traction. Things would have been even worse had it been command-line based!

We appreciate the comment here, but have not made steps to include the effects of compositional zoning in our software. This could perhaps be a feature to include in the future, but our focus was on exploring ranges of grain sizes, eU concentrations, and cooling rates so that the resulting age/closure temperature patterns could be considered in interpreting groups of ages. We have, however, noted in the text that this is another factor to consider in the interpretation of individual grain ages (see lines 84-86).

With a bit of extra work, you could turn this good paper into an excellent one. I will give you four weeks to address the points raised in this decision letter, but would be happy to extend this further if you need more time.

Thank you for submitting your work to GChron.

Non-public comments to the author:
Please do not hesitate to contact me directly should you have any questions.

We appreciate the feedback and suggestions provided from the reviewers on our Short Communication. Below we provide responses and indicate any resulting changes in the revised manuscript in red. Reviewer comments are in italicized text.

RC1: 'Comment on gchron-2021-29', William Guenthner, 23 Nov 2021

*Whipp and co-authors present a set of figures and accompanying code that illustrates the interaction among spherical radius, effective uranium (eU), and closure temperature for the apatite and zircon (U-Th)/He systems. This short communication will be a useful reference for illustrating how radiation damage (as proportional to eU here) influences closure temperature for a variety of grain sizes and cooling rates. This article is a nice contribution that provides several useful modeling tools and reference points for the community. I appreciate that someone has created plots of closure temperature for apatite and zircon (U-Th)/He data that are easily referenced and citable, and that go beyond the canonical closure temperature numbers that are still cited in some parts of the literature. The article could be essentially published as is given the scope of the short communication submission goals. I do have a few minor comments though:*

*Line 61: typo with "concentrations"*

Amended

*Line 91: Could you provide more details on the software here? Open-source code? What language? Basic design?*

We have modified and added a few sentences to give some additional information about the software, as well as more clearly refer the reader to where to find the software if they would like to use it. The modified text now reads: "The software used comprises programs for predicting (U-Th)/He (Ketcham et al., 2018) and apatite fission-track (Ketcham et al., 2000, as implemented in Braun et al., 2012) closure temperatures and ages written by Richard Ketcham in the C and C++ programming languages, and our scripts written in the Python language for producing the cooling histories and plots (each figure has a separate script file). The software is all open source and details about licensing can be found in the software archive online (see Code availability section). Each script file has a section of user-modifiable values at the start of the script and using this software it is possible for users to reproduce and customize versions of Figures 2-5. Furthermore, in addition to the linear cooling histories presented here it is possible to define more complex thermal histories involving multi-stage cooling and reheating events, as well as export predicted AFT length distributions. Details about how to use and customize the software are available in the code description in the software archive online."

*Line 103: provide a short definition of what ESR is and why it's a useful concept for thinking about grain size*

Response: We have added the following text to the manuscript: He mobility is a thermally-controlled volume diffusion process and therefore sensitive to grain size (e.g., Reiners and Farley, 2001), which is typically quantified as equivalent spherical radius (ESR), based on the observation that isothermal outgassing of apatite well fits a spherical diffusion model (Wolf et al., 1996).

*Line 115: also the annealing temperatures are higher than apatite*

Response: Agree. We have added the following text to the manuscript: "Zircon annealing temperatures are higher than for apatite, and zircon is also more resistant to geological cycling than apatite."

*Line 177: this is an interesting observation, and deserving of a little more explanation and highlighting. Do you an intuitive sense for why this is? Or, more precisely, can you explain this behavior in the context of the damage-diffusivity response?*

Response: This outcome surprised us also and we agree that it deserves more explanation in the context of damage-diffusivity in zircon. The observation is that the most pristine zircon (low Eu, fast cooling rate) diffuses He more efficiently than even slightly damaged zircon. Usually in thermal diffusion systems faster cooling rates result in higher effective closure temperatures. Yet, it has been shown (in a foundational contribution by the reviewer) that alpha damage in zircon decreases dramatically with increased alpha dosage for low magnitudes of alpha dosage, likely by increasing the tortuosity of the He diffusion pathways in the crystal (Guenthner et al., 2013 AJS, doi: 10.2475/03.2013.01). For a given cooling rate, crystals with low amounts of damage result in a higher effective closure temperature (and older cooling age) than crystals with no damage. What we observe here is that since the time accumulating damage is shorter under faster cooling scenarios, the decreased degree of damage in zircon for the faster cooled scenarios must a greater impact on closure T and cooling age than the faster cooling rate itself. We do not see this effect manifest in the apatite models (Fig. 3b). We have added the following text to the manuscript to discuss this: "Although this result seems potentially contrary to thermal diffusion systems in which faster cooling rates generally result in higher closure temperatures (e.g. Fig. 4b), in the case of zircon, the faster cooling scenarios also result in the most pristine crystals because of the increasingly shorter time to accumulate alpha damage (low eU, fast cooling rate). Moderate dosages of alpha damage are known to decrease diffusivity compared to pristine to low alpha damage dosage crystals (Guenthner et al., 2013). The implications is that zircon with low eU under fast cooling rates is more sensitive to the degree of alpha damage than to cooling rate."

*Figure 1: Because of the thermal history used here, the age differences in a and c are slight, and within the standard deviation that we typically see in apatite and zircon (U-Th)/He dates without any radiation damage effect. I'm wondering if a slower cooling rate and therefore longer time period might be more representative of a "typical" time-temperature path for which we'd want to investigate a date-eU correlation? Maybe all that changes with a longer time-temperature path is the absolute numbers, but the contours look relatively the same, so the basic point here remains unchanged. It just seems that a cooling history of only 25-0 Ma and 250-0 oC would not really yield a date-eU correlation in natural samples that would be interpretable.*

Response: We appreciate and fully agree with this comment and suggestion. In response, we have added a new Figure 3, which reproduces the ESR and eU conditions of Fig. 2, but applying a 1 C/myr cooling rate (model run time of 250 myr). To discuss the differences between Fig. 2 and Fig. 3 we have added the following text: "While the 10 °C/Myr cooling rate applied in Figure 2 could represent active orogenic settings, slower cooling rates are also common to many geological environments. To compare the effect of an order of magnitude slower cooling on He diffusion, we have applied all the same parameters as in the 10 °C/Myr scenario to a 1 °C/Myr constant cooling rate, equating to a model run time of 250 Myr (Fig. 3). The primary difference in the behavior of He in apatite under slower cooling is that AHe ages and closure temperatures correlate much more strongly with eU concentration than with ESR, resulting in ~30 °C variability in closure temperature over the range of eU concentration considered (Fig. 3b) and variation in AHe age of up to 30 Myr (Fig. 3a). The slower cooling provides a greater period

of time for accumulation of alpha decay-induced crystal damage. In an empirical study, such a cooling scenario could be expected to produce statistically-significant positive age-eU correlations. The ZHe system in this scenario continues to be insensitive to ESR, and while it is highly sensitive to eU concentration for values <500 ppm, it is quite insensitive to eU concentration for values >500 ppm (Fig. 3c,d). Thus, there is an eU concentration threshold above which zircons may not show an age-eU relationship. This threshold could be important to recognize when interpreting the significance of zircon age-eU plots."

RC2: 'Comment on gchron-2021-29', Christoph Glotzbach, 23 Nov 2021

**General comments:**

*Whipp et al. present explored the full range of low-temperature thermochronological closure temperatures as a function of uniform cooling rate, grain size and effective uranium. Their age predictions and derived closure temperatures are based on state-of-the-art diffusion and annealing models. They found a range of boundary conditions under which thermochronological ages overlap (e.g. apatite (U-Th)/He ages older than apatite fission track ages). Overall this short communication does present high-quality figures that can be used for teaching and making rough interpretation of multi-thermochronological data. The only substantial correction I ask the authors is a comparison of predicted ages (with their software) with those predicted by HeFTy (to make sure that they have integrated the provided annealing and diffusion models from Richard Ketcham correctly). After adding this, I am optimistic that the manuscript from Whipp et al. will be a valuable contribution for the thermochronological community.*

**Technical corrections:**

*Line 91-95: I would add a short paragraph and report some details of the software.*

We have modified and added a few sentences to give some additional information about the software, as well as more clearly refer the reader to where to find the software if they would like to use it. The modified text now reads: "The software used comprises programs for predicting (U-Th)/He (Ketcham et al., 2018) and apatite fission-track (Ketcham et al., 2000, as implemented in Braun et al., 2012) closure temperatures and ages written by Richard Ketcham in the C and C++ programming languages, and our scripts written in the Python language for producing the cooling histories and plots (each figure has a separate script file). The software is all open source and details about licensing can be found in the software archive online (see Code availability section). Each script file has a section of user-modifiable values at the start of the script and using this software it is possible for users to reproduce and customize versions of Figures 2-5. Furthermore, in addition to the linear cooling histories presented here it is possible to define more complex thermal histories involving multi-stage cooling and reheating events, as well as export predicted AFT length distributions. Details about how to use and customize the software are available in the code description in the software archive online."

*I also encourage the authors to provide information about the comparison with age predictions made in HeFTy. Simply model for the most extreme cases and some intermediate the corresponding thermochronological ages with HeFTy and your own software.*

Response: We appreciate this suggestion and since HeFTy is a widely used tool, this is an important point. We have added a direct comparison between HeFTy model predictions and our software, in which we model the four corners of Figs. 2, 3, and 4, and the fastest and slowest cooling scenarios in Fig. 5 (revised version figure numbering). We have used the same code used in HeFTy for the (U-Th)/He age prediction in our results, so the predicted ages vary by less than 0.8% from the values in HeFTy. The fission-track age prediction was taken from Pecube (Braun et al., 2012) using the code described in Ketcham et al. (2000). The comparison table is show below and will be added to the final software archive if the article is accepted.

| | Figure | mineral | eU (ppm) | ESR (um) | Start T (°C) | End T (°C) | run time (myr) | cooling rate (°C/myr) | HeFTy ZHe age (Ma) | This model ZHe age (Ma) | % difference | HeFTy AFT age (Ma) | This model AFT age (Ma) | % difference | HeFTy AHe age (Ma) | This model AHe age (Ma) | % difference |
|---|---|---|---|---|---|---|---|---|---|---|---|---|---|---|---|---|---|
| Run 1 | 2a | apatite | 1 | 40 | 350 | 0 | 35 | 10 | | | | | | | 5.56 | 5.56 | 0.00 |
| Run 2 | 2a | apatite | 1 | 100 | 350 | 0 | 35 | 10 | | | | | | | 6.83 | 6.83 | 0.00 |
| Run 3 | 2a | apatite | 150 | 40 | 350 | 0 | 35 | 10 | | | | | | | 6.09 | 6.09 | 0.00 |
| Run 4 | 2a | apatite | 150 | 100 | 350 | 0 | 35 | 10 | | | | | | | 7.13 | 7.13 | 0.00 |
| Run 5 | 2c | zircon | 1 | 40 | 350 | 0 | 35 | 10 | 7.35 | 7.35 | 0.00 | | | | | | |
| Run 6 | 2c | zircon | 1 | 100 | 350 | 0 | 35 | 10 | 8.38 | 8.38 | 0.00 | | | | | | |
| Run 7 | 2c | zircon | 4000 | 40 | 350 | 0 | 35 | 10 | 17.9 | 17.9 | 0.00 | | | | | | |
| Run 8 | 2c | zircon | 4000 | 100 | 350 | 0 | 35 | 10 | 19.5 | 19.5 | 0.00 | | | | | | |
| Run 9 | 3a | apatite | 1 | 40 | 350 | 0 | 350 | 1 | | | | | | | 40.3 | 40.3 | 0.00 |
| Run 10 | 3a | apatite | 1 | 100 | 350 | 0 | 350 | 1 | | | | | | | 51.7 | 51.7 | 0.00 |
| Run 11 | 3a | apatite | 150 | 40 | 350 | 0 | 350 | 1 | | | | | | | 69.6 | 69.6 | 0.00 |
| Run 12 | 3a | apatite | 150 | 100 | 350 | 0 | 350 | 1 | | | | | | | 75.8 | 75.8 | 0.00 |
| Run 13 | 3c | zircon | 1 | 40 | 350 | 0 | 350 | 1 | 81.0 | 81.0 | 0.00 | | | | | | |
| Run 14 | 3c | zircon | 1 | 100 | 350 | 0 | 350 | 1 | 91.0 | 91.0 | 0.00 | | | | | | |
| Run 15 | 3c | zircon | 4000 | 40 | 350 | 0 | 350 | 1 | 144.2 | 144.2 | 0.00 | | | | | | |
| Run 16 | 3c | zircon | 4000 | 100 | 350 | 0 | 350 | 1 | 166.1 | 166.1 | 0.00 | | | | | | |
| Run 17 | 4a | apatite | 10 | 40 | 350 | 0 | 3500 | 0.1 | | | | | | | 519.6 | 519.6 | 0.00 |
| Run 18 | 4a | apatite | 10 | 40 | 350 | 0 | 3.5 | 100 | | | | | | | 0.726 | 0.726 | 0.00 |
| Run 19 | 4a | apatite | 10 | 100 | 350 | 0 | 3500 | 0.1 | | | | | | | 578.8 | 578.7 | 0.02 |
| Run 20 | 4a | apatite | 10 | 100 | 350 | 0 | 3.5 | 100 | | | | | | | 0.866 | 0.866 | 0.00 |
| Run 21 | 4b | apatite | 1 | 45 | 350 | 0 | 3500 | 0.1 | | | | | | | 317.1 | 317.1 | 0.00 |
| Run 22 | 4b | apatite | 1 | 45 | 350 | 0 | 3.5 | 100 | | | | | | | 0.744 | 0.745 | -0.13 |
| Run 23 | 4b | apatite | 150 | 45 | 350 | 0 | 3500 | 0.1 | | | | | | | 763.0 | 763.2 | -0.03 |
| Run 24 | 4b | apatite | 150 | 45 | 350 | 0 | 3.5 | 100 | | | | | | | 0.745 | 0.745 | 0.00 |
| Run 25 | 4c | zircon | 100 | 40 | 350 | 0 | 3500 | 0.1 | 1459 | 1459 | 0.00 | | | | | | |
| Run 26 | 4c | zircon | 100 | 40 | 350 | 0 | 3.5 | 100 | 1.13 | 1.13 | 0.00 | | | | | | |
| Run 27 | 4c | zircon | 100 | 100 | 350 | 0 | 3500 | 0.1 | 1609 | 1609 | 0.00 | | | | | | |
| Run 28 | 4c | zircon | 100 | 100 | 350 | 0 | 3.5 | 100 | 1.24 | 1.24 | 0.00 | | | | | | |
| Run 29 | 4d | zircon | 1 | 60 | 350 | 0 | 3500 | 0.1 | 1017 | 1017 | 0.00 | | | | | | |
| Run 30 | 4d | zircon | 1 | 60 | 350 | 0 | 3.5 | 100 | 0.924 | 0.924 | 0.00 | | | | | | |
| Run 31 | 4d | zircon | 4000 | 60 | 350 | 0 | 3500 | 0.1 | 0.002 | 0.000 | #DIV/0! | | | | | | |
| Run 32 | 4d | zircon | 4000 | 60 | 350 | 0 | 3.5 | 100 | 1.74 | 1.74 | 0.00 | | | | | | |
| Run 33 | 5a | zircon/apatite | 10.0/1.0 | 60/45 | 350 | 0 | 3500 | 0.1 | 1313 | 1313 | 0.00 | 718 | 716 | 0.28 | 317.1 | 317.1 | 0.00 |
| Run 34 | 5a | zircon/apatite | 10.0/1.1 | 60/45 | 350 | 0 | 3.5 | 100 | 0.924 | 0.924 | 0.00 | 1.26 | 1.26 | 0.00 | 0.744 | 0.745 | -0.13 |
| Run 35 | 5c | zircon/apatite | 200.0/20.0 | 60/45 | 350 | 0 | 3500 | 0.1 | 1468 | 1468 | 0.00 | 718 | 716 | 0.28 | 594.9 | 595 | -0.02 |
| Run 36 | 5c | zircon/apatite | 200.0/20.0 | 60/45 | 350 | 0 | 3.5 | 100 | 1.27 | 1.28 | -0.78 | 1.26 | 1.26 | 0.00 | 0.744 | 0.745 | -0.13 |
| Run 37 | 5e | zircon/apatite | 4000/150.0 | 60/45 | 350 | 0 | 3500 | 0.1 | 0.002 | 0.000 | #DIV/0! | 718 | 716 | 0.28 | 763 | 763 | 0.00 |
| Run 38 | 5e | zircon/apatite | 4000/150.0 | 60/45 | 350 | 0 | 3.5 | 100 | 1.74 | 1.74 | 0.00 | 1.26 | 1.26 | 0.00 | 0.745 | 0.745 | 0.00 |